# Improving OOD Generalization of Pre-trained Encoders via Aligned Embedding-Space Ensembles

**Shuman Peng, Arash Khoeini, Sharan Vaswani, Martin Ester**
School of Computing Science
Simon Fraser University
{shumanp, akhoeini}@sfu.ca  vaswani.sharan@gmail.com  ester@cs.sfu.ca

## Abstract

The quality of self-supervised pre-trained embeddings on out-of-distribution (OOD) data is poor without fine-tuning. A straightforward and simple approach to improving the generalization of pre-trained representation to OOD data is the use of deep ensembles. However, obtaining an effective ensemble in the embedding space with only unlabeled data remains an unsolved problem. We first perform a theoretical analysis that reveals the relationship between individual hyperspherical embedding spaces in an ensemble. We then design a principled method to align these embedding spaces in an unsupervised manner. Experimental results on the MNIST dataset show that our embedding-space ensemble method improves pre-trained embedding quality on in-distribution and OOD data compared to single encoders.

## 1 Introduction

Self-supervised learning techniques enable the pre-training of deep neural network (DNN) encoders on widely available unlabeled data. These pre-trained encoders, once fine-tuned, are highly transferable to various downstream tasks [5, 3]. However, a key challenge remains: without fine-tuning, the quality of pre-trained features is noticeably lower on out-of-distribution (OOD) data, impairing the performance of pre-trained models on subsequent OOD downstream tasks [7]. This issue is particularly critical when the downstream task has insufficient data for fine-tuning, making it essential for the pre-trained encoders to generalize well to OOD data in a zero-shot manner without fine-tuning.

A straightforward approach to improve the generalizability of pre-trained representation quality to OOD data is the use of deep ensembles. Deep ensembles (DEs) [9], consist of DNNs independently trained with different initializations and data orders (e.g. seeds). DEs have been shown to improve the predictive performance over single DNNs on both in-distribution (ID) and OOD data [15, 1].

Existing DE approaches typically aggregate models either in the predictive output space [1, 9, 15] (e.g., logits for classification) or in the weight space [22, 16, 17]. Aggregating in the predictive space confines the ensemble to a single task, e.g., classification with a fixed set of classes, and is inapplicable for self-supervised pre-trained encoders. In contrast, aggregating models in the weight space offers the flexibility to accommodate various tasks by discarding the predictive layer and retaining only the ensembled encoder. However, this approach sacrifices the interpretability that predictive-space ensembles provide, particularly in how individual model outputs are combined and unified. In essence, existing DE techniques exhibit an undesirable trade-off between interpretability and flexibility.

In this paper, we take a novel perspective to ensemble self-supervised pre-trained encoders for improved zero-shot OOD generalization. Our approach offers both the flexibility of weight-space ensembling and the interpretability of predictive-space ensembling. We propose an embedding-space ensemble, called `Ensemble-InfoNCE`, in which we aggregate the ensemble mean in the

NeurIPS 2024 Workshop on Unifying Representations in Neural Models (UniReps 2024).

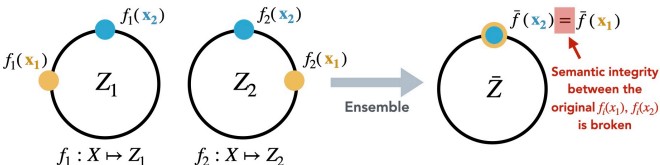

Figure 1: Need for embedding alignment: The ensemble mean of two different embeddings (yellow, blue) in misaligned embedding spaces $Z_1, Z_2$ collapses to the same vector in $\bar{Z}$, although they have different semantic meanings.

hyperspherical latent embedding space of encoders pre-trained using the widely used InfoNCE contrastive loss [14, 3].

Obtaining the ensemble mean of embedding vectors is less straightforward compared to taking the mean of predictive outputs or model weights. To take the ensemble mean of embeddings, the embedding spaces must be aligned such that the embeddings (produced by the different encoders) corresponding to semantically similar samples have a similar direction in the hyperspherical space. Taking the mean of misaligned embeddings that point in different directions can harm the semantic integrity of the embedding space (Figure 1). Existing approaches align embeddings using class labels [23]. However, aligning embeddings without access to labels remains an unsolved problem. To this end, we extend the theoretical results in [25], and use this to propose a principled unsupervised approach (referred to as `Ensemble-InfoNCE`) to align the embeddings. Our theoretical results demonstrate that an ensemble of encoders with aligned embedding spaces recovers the correct (ground truth) embeddings. Finally, we experimentally show improved pre-trained embedding quality on in-distribution (ID) and OOD data for `Ensemble-InfoNCE` compared to single InfoNCE encoders on the MNIST dataset.

## 2 Background and preliminaries

**InfoNCE loss.** We consider encoders pre-trained using the InfoNCE loss, a widely used contrastive (self-supervised) learning objective [14, 2, 3, 5, 19, 21]. These encoders $f$ map the input space $\mathcal{X}$ to an $L_2$-normalized unit-hyperspherical embedding space $\mathcal{Z} = \mathbb{S}^{D-1}$. Encoders trained with the InfoNCE loss (3) have the desirable property of mapping semantically similar pairs of samples close together in the embedding space, while also ensuring that dissimilar samples are mapped far apart.

**InfoNCE-trained encoders recover the correct latents.** Zimmermann et al. [25] theoretically demonstrate that an encoder $f$ minizing the contrastive loss in (3) recovers the ground truth latents $z \in \mathcal{Z}$ up to orthogonal transformations. Specifically, $z_1^\top z_2 = h(z_1)^\top h(z_2)$, where $z_1, z_2$ are ground truth latents, and $h = f \circ g$ composes of encoder $f$ and a generative process $g : \mathcal{Z} \mapsto \mathcal{X}$.[1]

## 3 Embedding-space ensembles via unsupervised alignment

In this section, we introduce, to the best of our knowledge, the first approach of ensembling self-supervised pre-trained encoders in the embedding space, which offers the interpretability of predictive-space ensembles and the flexibility of weight-space ensembles. Our embedding-space ensemble approach, referred to as `Ensemble-InfoNCE`, produces mean embedding vectors $\bar{z} = \bar{f}(x) \in \mathcal{Z}$ for a given ensemble of $M > 1$ encoders $\{f_i : \mathcal{X} \mapsto \mathcal{Z}_i\}_{i=1}^M$ pre-trained using the InfoNCE loss (3). Each $\mathcal{Z}_i$ is a unit-hyperspherical embedding space $\mathcal{Z}_i = \mathbb{S}^{D-1}$. Before taking the ensemble mean, we must first *align* the embedding spaces $\{\mathcal{Z}_i\}_{i=1}^M$ so that each of the $M$ embeddings $\{f_i(x)\}_{i=1}^M$ for the same input $x$ points in a similar direction on a hypersphere. However, aligning embedding spaces without labels in an unsupervised manner remains a challenging problem.

To tackle the challenge of performing unsupervised embedding space alignment, we first conduct a theoretical analysis and reveal a critical orthogonality relationship between different embedding spaces (Section 3.1). This relationship allows us to extend the guarantees on the recovery of the correct latents[2] from single encoders to an ensemble. Furthermore, this relationship enables us to align the embedding spaces by recovering the orthogonal transformation matrix and forms the basis of our unsupervised embedding space alignment approach (Section 3.2). Finally, the aligned ensemble embeddings are aggregated using the Karcher Mean algorithm [18] (Section 3.3).

---

[1]We refer interested readers to Theorem 2 of [25].

[2]In this paper, we use the terms "latents", "features", and "embeddings" interchangeably.

## 3.1 Theoretical analysis

**An encoder's latent space is an orthogonal transformation of another encoder's.** Extending the main theoretical result of [25] (Section 2) from single encoders to an ensemble of these encoders, we reveal that an encoder's embedding space is an orthogonal transformation of another encoder's. This is formally stated below in Proposition 1 with the corresponding proof in Appendix A.2.

**Assumption.** For an ensemble of encoders $f_1, f_2$ trained on the same data $D = \{x_i\}_{i=1}^N$ with different random seeds, we *assume* that $f_1(x) = R_1 z$ and $f_2(x) = R_2 z$. In other words, both $f_1$ and $f_2$ recover the correct (ground truth) latents $z$ up to different orthogonal transformations $R_1, R_2$.[3]

**Proposition 1** (Orthogonal transformation relationship)**.** *Under the above assumption, $f_1$ and $f_2$ learn the same latents up to an orthogonal transformation $R$, that is, $f_1(x) = R f_2(x)$.*

**An ensemble of aligned embeddings recovers the correct latents.** Based on Proposition 1, in Proposition 2 we generalize the theoretical guarantee on the correctness of the learned latents for each ensemble member to the ensemble as a whole. The proof is provided in Appendix A.2.

**Proposition 2** (Ensemble recovers correct latents)**.** *The ensemble mean $\bar{f}(x)$ of aligned embeddings $f_1(x)$ and $R f_2(x)$ are the correct latents $z$ up to orthogonal transformation $R_1$, that is, $\bar{f}(x) = R_1 z$.*

**Approximate orthogonal relationship on real-world data.** With real-world data, there may not exist an orthogonal transformation that perfectly aligns *all* corresponding embedding vectors of the same input between different embedding spaces. This discrepancy arises due to violations of the data generation and modeling assumptions. Under such violated conditions, Zimmermann et al. [25] demonstrated that these encoders still recover the true latents to a moderate to high degree. By extending this result to our problem setting, we infer that the relationship between encoders $f_i$ and $f_j$ can be approximated as $f_i(x) \approx R f_j(x)$ for all $x \in \mathcal{X}$. In deep ensembles, this approximate orthogonal transformation relationship preserves some degree of diversity between embedding spaces, preventing the ensemble from collapsing into a single model.

## 3.2 Unsupervised embedding space alignment via learning orthogonal matrices

Our goal is to align the $M$ hyperspherical embedding spaces $Z_1, ..., Z_M$ induced by encoders $\{f_i\}_{i=1}^M$ so that the same in-distribution pre-training sample is mapped to embeddings that have a similar direction across $\{Z_i\}_{i=1}^M$. From Section 3.1, we know that $f_i(x) \approx R f_j(x)$ for all $x \in \mathcal{X}$, which means that learning the orthogonal transformation matrix $R$ would naturally align $f_j(x)$ with $f_i(x)$. We note that $R$ does not need to be strictly orthogonal for aligning the embedding spaces. To learn $R$ in a $D$-dimensional embedding space, we use a single-layer neural network with $D$ input and output nodes. The $D \times D$ weight matrix within this single layer neural network represents the orthogonal transformation matrix $R$ that we want to learn.

To align the embedding spaces, we randomly select one **anchor** encoder from the set of $M$ encoders to align the remaining $M - 1$ encoders. To align each $Z_j$ with the anchor embedding space $Z_i$, we propose an objective function that enforces $R$ to be as close to orthogonal as possible by imposing orthogonality as a soft constraint, while simultaneously maximizing the alignment between pairs of embeddings:

$$\mathcal{L}_{\text{align}} = \underset{R \in \mathbb{R}^{D \times D}}{\arg\min} \frac{1}{N} \sum_{n=1}^N d(f_i(x_n), R f_j(x_n)) + \lambda \|R^T R - I_D\|_F^2 \tag{1}$$

where $N$ denotes the number of samples, $\lambda$ is a hyperparameter that controls the strength of the orthogonality constraint, and $d(.,.)$ is a function that quantifies the discrepancy between pairs of vectors. Given that the embedding vectors reside on the surface of a unit sphere, we use the geodesic distance, which measures the shortest path between two points on a Riemannian manifold and accounts for the spherical geometry. The objective function then becomes:

$$\mathcal{L}_{\text{align}} = \underset{R \in \mathbb{R}^{D \times D}}{\arg\min} \frac{1}{N} \sum_{n=1}^N \arccos(f_i(x_n), R f_j(x_n)) + \lambda \|R^T R - I_D\|_F^2 \tag{2}$$

---

[3]For our theoretical analysis, we consider the case of an ensemble of $M = 2$ encoders for simplicity. The results can be extended to $M > 2$ encoders.

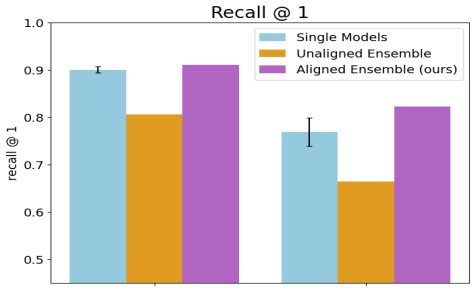
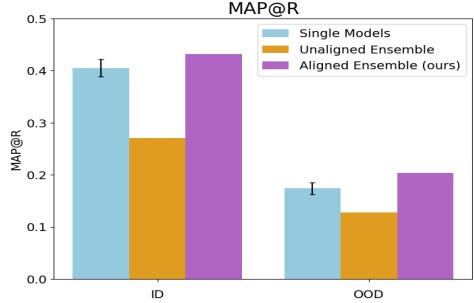

Figure 2: Comparing embedding qualities of single models (blue), an ensemble of unaligned embedding spaces (orange), and an ensemble of aligned embedding spaces (purple) in the ID and OOD settings. Recall@1 and MAP@R are presented. Higher values indicate better performance. The mean and standard deviation (error bars) of the performance metrics are reported for the 5 single models. The ensembles do not have standard deviation since all 5 models are combined into one.

### 3.3 Embedding mean in an ensemble

An ensemble of $M$ pre-trained encoders with aligned embedding spaces produces a mean embedding vector for each sample, i.e., $\bar{z} = \bar{f}(x) = \text{mean}(f_1(x), R_2 f_2(x), ..., R_M f_M(x))$. Given the hyperspherical nature of the embedding spaces, we apply the Karcher Mean algorithm from [18] to compute meaningful mean vectors on the surface of a sphere. The algorithm projects hyperspherical data points onto a linear tangent space, calculates the mean in this tangent space, and then projects the result back onto the sphere. This process iterates until the mean in the tangent space approaches a near-zero norm, indicating that the mean has converged.[4]

## 4 Experiments

**Dataset and training setup**    We use the MNIST dataset [10]. For ID evaluation, the test set is used as is. For OOD evaluation, each test sample is randomly colored. Since random coloring was not applied during pre-training, colored versions of the images are considered OOD. A total of $M = 5$ encoders are trained for the ensemble, as this ensemble size has been shown to be sufficient to produce good results [15]. Details of the dataset, model architecture and training are provided in Appendix A.3 and A.4.

**Evaluation metrics**    In line with the representation learning literature, we assess the quality of embeddings using the **Recall at 1** (R@1) and **Mean Average Precision at R** (MAP@R) metrics [7, 12]. R@1 measures the semantic quality of the embedding space by verifying if each embedding's nearest neighbor belongs to the same class. MAP@R evaluates the proportion of each embedding's $R$ nearest neighbors that belong to the same class, while accounting for the ranking of correct retrievals [12]. $R$ is set to the total number of samples in a class [12].

### 4.1 Experimental results

Figure 2 shows that our `Ensemble-InfoNCE` model with aligned embedding spaces (shown in purple) improves the quality of embeddings over single models and unaligned ensembles in both the in-distribution (ID) and out-of-distribution (OOD) settings. The embedding quality improvement achieved by our method is more pronounced in the OOD setting, with a $6.99\%$ improvement over the mean R@1 of the $M = 5$ single models and a $17.38\%$ improvement over the mean MAP@R of single models. Our results also highlight the importance of aligned embedding spaces for ensembles. Taking an ensemble of misaligned embedding spaces consistently hurts the embedding quality, achieving lower values of R@1 and MAP@R than a single model for both ID and OOD settings. Table 1 provides the numerical values for Figure 2, and additional results are provided in Appendix A.5.

## 5 Conclusion

We improved the generalizability of pre-trained encoders to OOD data by taking an ensemble of encoders in the embedding space. We constructed embedding-space ensembles by effectively aligning the individual embedding spaces. Preliminary experiments on the MNIST dataset demonstrate that our aligned embedding-space ensemble significantly enhances the OOD embedding quality compared to individual models. In the future, we will focus on scaling our method to larger datasets, such as ImageNet, and incorporating OOD data from real datasets [4, 8, 20, 13].

---

[4]Details of the Karcher Mean algorithm can be found in [18].

## Acknowledgments and Disclosure of Funding

This research was supported by the NSERC Discovery Grant. We would like to thank Dr. Ke Li for his early feedback. We are grateful to Shichong Peng for providing insightful discussions and thorough feedback throughout the course of this research. We also appreciate the anonymous reviewers for their constructive feedback and suggestions.

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
