# Improving OOD Generalization of Pre-trained Encoders via Aligned Embedding-Space Ensembles

**Shuman Peng, Arash Khoeini, Sharan Vaswani, Martin Ester**
School of Computing Science
Simon Fraser University
{shumanp, akhoeini}@sfu.ca  vaswani.sharan@gmail.com  ester@cs.sfu.ca

## A    Appendix / supplemental material

### A.1    The InfoNCE contrastive loss

The InfoNCE loss is defined as

$$
L_{\text{contr}}(f; \tau, M) \quad := \quad \mathop{\mathbb{E}}_{\substack{(x,\, x^+) \sim p_{\text{pos}} \\ \{x_i^-\}_{i=1}^M \overset{\text{i.i.d.}}{\sim} p_{\text{data}}}} \left[ -\log \frac{e^{f(x)^\mathsf{T} f(x^+)/\tau}}{e^{f(x)^\mathsf{T} f(x^+)/\tau} + \sum_{i=1}^M e^{f(x)^\mathsf{T} f(x_i^-)/\tau}} \right], \tag{3}
$$

where pairs $(x, x^+)$ are drawn of the distribution of positive samples $p_{\text{pos}}$, and $M > 0$ negative samples are drawn from the distribution of all observations $p_{\text{data}}$.

### A.2    Proofs for Section 3.1

**Proposition 1**  *Under the above assumption, $f_1$ and $f_2$ learn the same latents up to an orthogonal transformation R, that is, $f_1(x) = R f_2(x)$.*

*Proof.* Let $f_1(x) = R_1 z$ and $f_2(x) = R_2 z$ where $R_1, R_2$ are orthogonal matrices, i.e., $R_1^T = R_1^{-1}, R_2^T = R_2^{-1}$, we have

$$
\begin{aligned}
&\Rightarrow R_1^{-1} f_1(x) = z \quad \text{and} \quad R_2^{-1} f_2(x) = z \\
&\Rightarrow R_1^{-1} f_1(x) = R_2^{-1} f_2(x) \\
&\Rightarrow R_1 R_1^{-1} f_1(x) = R_1 R_2^{-1} f_2(x) \\
&\Rightarrow f_1(x) = R f_2(x) \quad \text{(Letting } R = R_1 R_2^{-1})
\end{aligned}
$$

Since $R_1$ and $R_2^{-1}$ are both orthogonal matrices, their product $R$ is also an orthogonal matrix, i.e., $R^T = R^{-1}$. Therefore, $f_1$ and $f_2$ learns the same latents up to an orthogonal transformation $R$.  □

**Proposition 2**  (The ensemble also recovers the correct latents). *The ensemble mean $\bar{f}(x)$ of aligned embeddings $f_1(x)$ and $R f_2(x)$ are the correct latents $z$ up to orthogonal rotation $R_1$, that is, $\bar{f}(x) = R_1 z$.*

*Proof.* Let us denote the ensemble mean as $\bar{f}(x) = \text{mean}(f_1(x), R f_2(x))$, where mean(.) is a general notion of the mean, which can be the arithmetic mean in Euclidean spaces or the Karcher Mean in Reimmanian manifolds. For the simplicity of this proof, we will use the arithmetic mean, but the results also apply to the Karcher Mean.

Defining $\bar{f}(x)$ using the arithmetic mean, we have:

NeurIPS 2024 Workshop on Unifying Representations in Neural Models (UniReps 2024).

$$\bar{f}(x) = \frac{1}{2}[f_1(x) + Rf_2(x)] \tag{4}$$

Since $f_1(x) = Rf_2(x)$, we have:

$$\bar{f}(x) = \frac{1}{2}[f_1(x) + f_1(x)] = f_1(x) = R_1 z \tag{5}$$

$\square$

### A.3 Additional implementation and training details

**Contrastive pre-training architecture** For contrastive pre-trained encoders, two convolution blocks with max-pooling and ReLU activations are used, with a linear layer attached at the end to project the embeddings down to $D = 8$ dimensions. The first convolution block consists of (1) a Conv2d with 3 input channels, 16 output channels, a kernel size of 5, stride of 1, and padding of 2; (2) max-pooling with kernel size 2; (3) a ReLU activation; and finally (4) a dropout layer with dropout rate $p$ (we used $p = 0.25$ in our experiments). Similarly, the second convolution block consists of identical components, except with a Conv2d that consists of 16 input channels and 32 output channels.

**Supervised contrastive pre-training** For part of our experiments (Figures 4 and 5), we follow [1] and **use class labels to generate positive and negative pairs**, which better preserves the theoretical assumptions and guarantees for encoders trained with the InfoNCE loss [6]. Positive pairs consist of samples from the same class, while negative pairs are from different classes. We refer to this approach as *supervised contrastive pre-training*. Each encoder is contrastive trained for 2000 epochs using different random seeds $(10, 11, 12, 13, 14)$ and weight initializations. A batch size of 128 is used, and each sample is paired with 16 negative samples, following [1]. The learning rate is set to $0.01$ and the AdamW optimizer [3] is used.

**Unsupervised contrastive pre-training** For unsupervised contrastive pre-training, which is the conventional contrastive pre-training approach (also used for Figures 2 and 6 and Table 1), we **apply random rotations ($\pm 30$ degrees) to generate positive pairs**. Positive pairs are created by taking two randomly rotated views of the same sample, while negative pairs are formed from randomly rotated views of different samples. We also experimented with using random cropping to generate positive pairs, but found that this resulted in unstable model training. This instability is likely due to the nature of the MNIST images, which consist of white digits on a black background. Images that are cropped to include too much background and insufficient detail of the digit can result in distinct samples being mistakenly identified as similar. Each encoder is trained for 20 epochs using different random seeds $(10, 11, 12, 13, 14)$ and weight initializations. We use a batch size of 1024 and a learning rate of 0.1 with the LAMB optimizer [5], which is better suited for larger batch sizes.

**Unsupervised embedding space alignment** For unsupervised embedding space alignment, a linear layer with $D$ input and output dimensions is used. To align the **supervised contrastive pre-trained encoders**, the linear alignment layer layer is trained for 20 epochs with a learning rate of $0.1$, and an orthogonality regularization factor $\lambda = 0.5$ is applied. To align the **unsupervised contrastive pre-trained encoders**, which exhibit lower degrees of orthogonality compared to the supervised ones (as expected due to the violation of theoretical assumptions regarding the conditional distribution used to generate positive pairs [6]), the linear layer is trained for 20 epochs with a learning rate of 0.1 and $\lambda \in \{0.1, 0.3, 0.5\}$ was applied. A lower $\lambda$ relaxes the orthogonality constraint for encoders that have slightly weaker orthogonal relationships. In both cases, the linear layer weights were optimized using stochastic gradient descent (SGD).

**Computing resources** We used a single RTX 3090 GPU for our experiments. For MNIST scale experiments, any GPU with more than 8GB of VRAM would be sufficient.

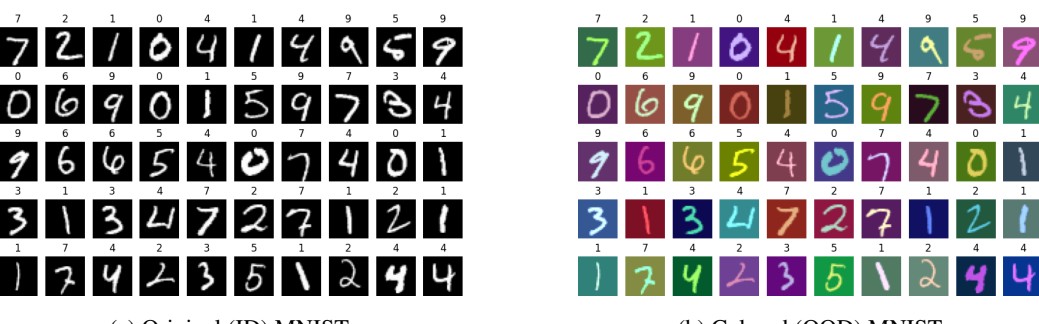

(a) Original (ID) MNIST                    (b) Colored (OOD) MNIST

Figure 3: For in-distribution (ID) evaluation, images like those in (a) were used. For out-of-distribution (OOD) evaluation, images like those in (b) were used.

## A.4 Additional dataset details

We converted single-channel grayscale MNIST images to three-channel black-and-white images. The training set is used to perform contrastive pre-training of the encoders and to align the embedding spaces. The test set is used for ID and OOD evaluation of the pre-trained encoders.

### A.4.1 Data for OOD evaluation

**Colored version** We randomly colored the images in the MNIST test set to create an OOD evaluation set. Since only black-and-white images were used during pre-training, colored versions of the images are considered OOD compared to the original images. Colored versions of the images are illustrated in Figure 3b.

**Cropped version** For further OOD evaluation, each test sample is randomly cropped to `crop_size` $\sim$ Unif($[0.25, 1]$) percent of their original size, following [1, 2]. Since no random cropping was applied during pre-training, cropped versions of the images are considered OOD compared to the original images.

## A.5 Additional results

### A.5.1 Supervised contrastive pre-training with colored OOD images

Figure 4 compares the embedding qualities of single models and ensembles of aligned and unaligned embedding spaces. The $M = 5$ models are trained using the supervised contrastive pre-training procedure discussed in Appendix A.3. OOD evaluation is performed on Colored MNIST images (Appendix A.4.1).

### A.5.2 Supervised contrastive pre-training with cropped OOD images

Figure 5 compares the embedding qualities of single models and ensembles of aligned and unaligned embedding spaces. The $M = 5$ models are trained using the supervised contrastive pre-training procedure discussed in Appendix A.3. Cropped MNIST (A.4.1) images are used as OOD evaluation data.

### A.5.3 Unsupervised contrastive pre-training with cropped OOD images

Figure 6 compares the embedding qualities of single models and ensembles of aligned and unaligned embedding spaces. The $M = 5$ models are trained using the unsupervised contrastive pre-training procedure discussed in A.3. Cropped MNIST (A.4.1) images are used as OOD evaluation data.

### A.5.4 Comparing embedding-space ensemble with weight-space ensemble

**Weight-space ensemble baselines** We create two types of weight-space ensembles: **weight-space ensemble (WSE)** and **weight-space ensemble\* (WSE\*)**. For WSE, we combine the $M$

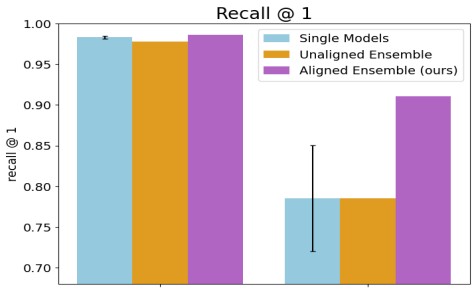 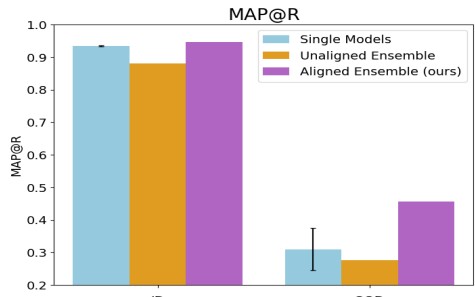

Figure 4: Supervised contrastive pre-training with Colored MNIST as OOD evaluation data. Comparing embedding qualities of single models (blue), an ensemble of unaligned embedding spaces (orange), and an ensemble of aligned embedding spaces (purple) in the ID and OOD settings. Recall@1 and MAP@R are presented. Higher values indicate better performance. The mean and standard deviation (error bars) of the performance metrics are reported for the 5 single models. The ensembles do not have standard deviation since all 5 models are combined into one.

independently contrastive pre-trained encoders – each with different initializations but otherwise identical hyperparameters – by taking the uniform mean of their weights. WSE is directly comparable to our embedding-space ensembles, as it uses the same set of single models. However, effective weight averaging requires that the models be trained from the same initialization, but with different hyperparameters sampled from a *mild search space*, which were only provided for ResNet50, to ensure the weights are averageable [4]. To satisfy these conditions for WSE*, we follow a similar approach by weight-averaging single models trained with carefully selected hyperparameters. To train the single MNIST models (non-ResNet50 architecture) we adjust the learning rate from $0.1$ by adding values in $\{0.00001, 0.00003, 0.00005\}$, and randomly sample the dropout rate from $\{0.25, 0.3\}$. Finally, WSE* is formed by taking the uniform mean of the resulting $M$ models.

**Results** Figures 7 and 8 compare the embedding quality across single models, unaligned embedding-space ensembles, aligned embedding-space ensembles, and both versions of weight-space ensemble discussed earlier. The straightforward weight-space ensemble approach (WSE), which uses the same single models as our embedding-space ensembles, consistently underperforms compared to the individual models, with performance reductions ranging from $36.13\%$ to $66.42\%$. The more carefully constructed weight-space ensemble (WSE*), which follows the standard weight-averaging practice, achieves performance comparable to the single models and consistently underperforms our aligned embedding-space ensemble in both ID and OOD settings. Detailed numerical values are provided in Tables 1 and 2.

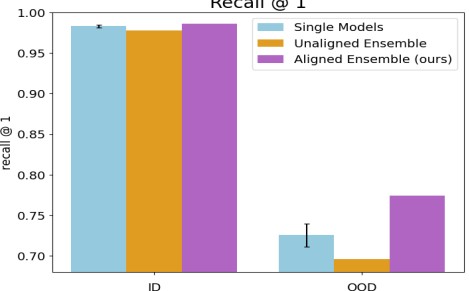 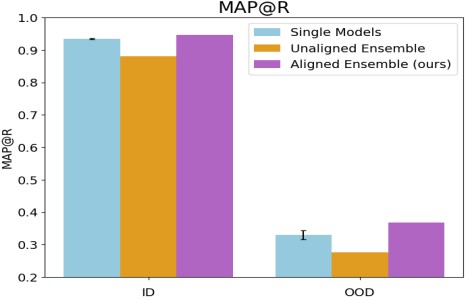

Figure 5: Supervised contrastive pre-training with Cropped MNIST as OOD evaluation data. Comparing embedding qualities of single models (blue), an ensemble of unaligned embedding spaces (orange), and an ensemble of aligned embedding spaces (purple) in the ID and OOD settings. Recall@1 and MAP@R are presented. Higher values indicate better performance. The mean and standard deviation (error bars) of the performance metrics are reported for the 5 single models. The ensembles do not have standard deviation since all 5 models are combined into one.

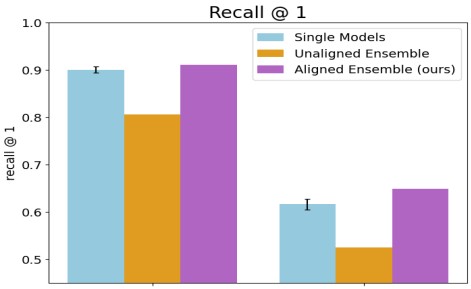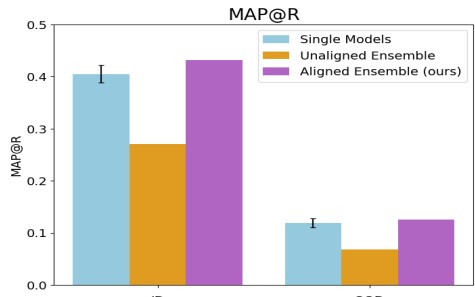

Figure 6: Unsupervised contrastive pre-training with Cropped MNIST as OOD evaluation data. Comparing embedding qualities of single models (blue), an ensemble of unaligned embedding spaces (orange), and an ensemble of aligned embedding spaces (purple) in the ID and OOD settings. Recall@1 and MAP@R are presented. Higher values indicate better performance. The mean and standard deviation (error bars) of the performance metrics are reported for the 5 single models. The ensembles do not have standard deviation since all 5 models are combined into one.

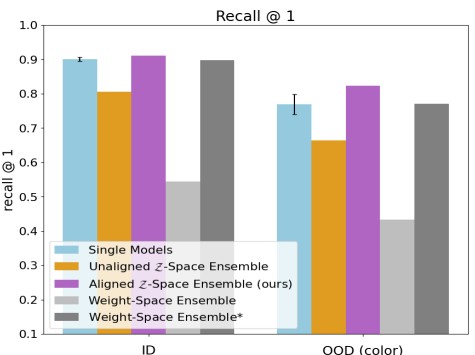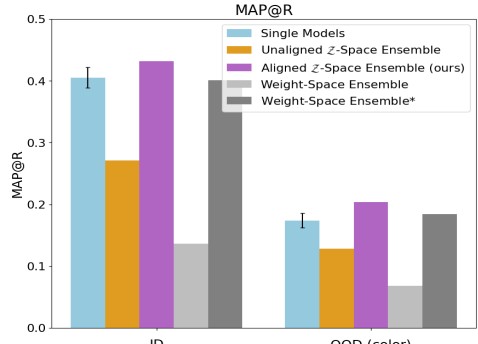

Figure 7: Comparing embedding qualities of single models (blue), an ensemble of unaligned embedding spaces (orange), an ensemble of aligned embedding spaces (purple), a straightforward weight-space ensemble (WSE) (light grey), and a more constructed crafted weight-space ensemble (WSE*) [4] (dark grey) in the ID and OOD settings. Recall@1 and MAP@R are presented. Higher values indicate better performance. The mean and standard deviation (error bars) of the performance metrics are reported for the 5 single models. The ensembles do not have standard deviation since all 5 models are combined into one. These plots show the same results as Figure 2, but with the addition of the weight-space ensemble baselines.

Table 1: Comparison of the embedding qualities of $M = 5$ **single models**, unaligned embedding-space ensemble (**Unaligned Encoders**), and aligned embedding-space ensemble (**Aligned Encoders**) in both in-distribution (ID) and out-of-distribution (OOD) settings. The mean and standard deviation of performance metrics (Recall@1, MAP@R) are reported for the single models. Standard deviation is not shown for the ensembles since all 5 models are combined into one. The top three rows represent Recall@1 performance, and the bottom three rows represent MAP@R performance. The $\%\Delta$ column shows the percentage change in performance for each ensemble type relative to single models. All models are trained on the MNIST dataset using the InfoNCE contrastive loss, where positive pairs are created by applying random rotations to the same input image.

| | | **Single Models** Mean $\pm$ Std | **Unaligned Encoders** Ensemble | $\%\Delta$ | **Aligned Encoders** Ensemble | $\%\Delta$ |
|---|---|---|---|---|---|---|
| Recall@1 ($\uparrow$) | ID | $0.900 \pm 0.006$ | 0.806 | -10.48% | **0.911** | +1.18% |
| | OOD (Color) | $0.769 \pm 0.029$ | 0.664 | -13.68% | **0.823** | +6.99% |
| | OOD (Crop) | $0.616 \pm 0.011$ | 0.525 | -14.80% | **0.649** | +5.32% |
| MAP@R ($\uparrow$) | ID | $0.405 \pm 0.017$ | 0.271 | -33.12% | **0.432** | +6.61% |
| | OOD (Color) | $0.174 \pm 0.012$ | 0.128 | -26.35% | **0.204** | +17.38% |
| | OOD (Crop) | $0.119 \pm 0.009$ | 0.068 | -42.95% | **0.126** | +5.71% |

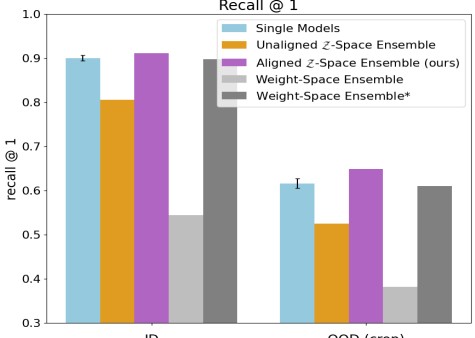 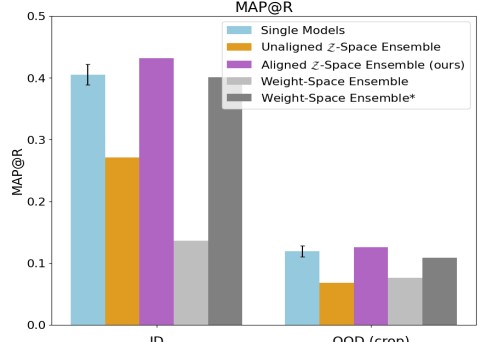

Figure 8: Comparing embedding qualities of single models (blue), an ensemble of unaligned embedding spaces (orange), an ensemble of aligned embedding spaces (purple), a straightforward weight-space ensemble (WSE) (light grey), and a more constructed crafted weight-space ensemble (WSE*) [4] (dark grey) in the ID and OOD settings. Recall@1 and MAP@R are presented. Higher values indicate better performance. The mean and standard deviation (error bars) of the performance metrics are reported for the 5 single models. The ensembles do not have standard deviation since all 5 models are combined into one. These plots show the same results as Figure 6, but with the addition of the weight-space ensemble baselines.

Table 2: Comparing the embedding qualities of $M = 5$ **single models**, straight-forward weight-space ensemble (**WSE**), and carefully constructed weight-space ensemble (**WSE***) in the in-distribution (ID) and out-of-distribution (OOD) settings. The mean and standard deviation of the performance metrics (Recall@1, MAP@R) are reported for the single models. The ensembles do not have standard deviation since all 5 models are combined into one. Entries in the top three rows represent the Recall@1 performance, and entries in the bottom three rows represent the MAP@R performance. The $\%\Delta$ column shows the percentage change in the respective ensemble type compared to single models. The models are trained on the MNIST dataset using the InfoNCE contrastive loss, where positive pairs are created by applying random rotations to the same input image.

|  |  | **Single Models** | **WSE** | | **WSE*** | |
|  |  | Mean $\pm$ Std | Ensemble | $\%\ \Delta$ | Ensemble | $\%\ \Delta$ |
|---|---|---|---|---|---|---|
| Recall@1 ($\uparrow$) | ID | $0.900 \pm 0.006$ | 0.544 | -39.56% | 0.898 | -0.22% |
|  | OOD (Color) | $0.769 \pm 0.029$ | 0.433 | -43.69% | 0.771 | +0.26% |
|  | OOD (Crop) | $0.616 \pm 0.011$ | 0.382 | -37.99% | 0.610 | -0.97% |
| MAP@R ($\uparrow$) | ID | $0.405 \pm 0.017$ | 0.136 | -66.42% | 0.401 | -0.99% |
|  | OOD (Color) | $0.174 \pm 0.012$ | 0.068 | -60.92% | 0.184 | +5.75% |
|  | OOD (Crop) | $0.119 \pm 0.009$ | 0.076 | -36.13% | 0.109 | -8.40% |