# OpenReview forum: "Improving OOD Generalization of Pre-trained Encoders via Aligned Embedding-Space Ensembles"
_NeurIPS.cc/2024/Workshop/UniReps — UniReps_

### Official Review · Reviewer_Edyv · 2024-10-03
**The proposed method is well-motivated and sounded.**

**Rating:** 6
**Confidence:** 4

**Review:**

The paper presents a approach to improve the OOD generalization of pre-trained encoders by leveraging ensemble learning in the embedding space. The results show that this method enhances the quality of embeddings for both ID and OOD data. Looking forward to the authors expanding the method to larger datasets in future works. Here are some issues:

1.The method claims to retain interpretability, more examples of how this is achieved would help strengthen the claim. I think providing visual or qualitative analysis of the aligned embeddings would be beneficial.

2.It does not include comparisons with more recent or relevant OOD generalization techniques.

3.Given the computational cost of ensembling and aligning embeddings, it would be useful to address how this method scales in terms of memory and time complexity.

---

### Official Review · Reviewer_nzmX · 2024-10-04
**This paper presents Ensemble-InfoNCE, a novel method for creating embedding-space ensembles of self-supervised models to improve OOD generalization. The approach aligns and aggregates embeddings from multiple InfoNCE-trained encoders. While theoretically grounded and showing promise on MNIST, the paper raises questions about computational scalability and sensitivity to imperfect alignments. Clarification on data augmentation during pre-training and more detailed analysis of the method's robustness would strengthen the work.**

**Rating:** 7
**Confidence:** 3

**Review:**

Summary: The authors provide a novel way of embedding-space ensemble, called Ensemble-InfoNCE, in which they aggregate the aligned ensemble mean of data in the hyperspherical latent embedding space of encoders pre-trained using InfoNCE contrastive loss. They provide the theoretical underpinning and a method to find the near orthogonal transformation, and show that the aligned mean vector performed better for the evaluation metrics for the embedding than single models in zero-shot learning.

Minor points:
- Lines 131-132: "since no image augmentation was applied during pre-training" It is unclear here what exactly is meant by "no image augmentation", and how the authors created positive and negative pairs for InfoNCE without augmentations (if that's indeed what they did). A clarification would be helpful.
- MAP@R: The sample size R used isn't given in the paper.

Major points:
- The extension of the theory from [22] seems straightforward and the authors' approach seems principled too. However, as such transformation is done on the embedding space for each model, it makes one curious how sensitive the method is to imperfect alignments between embedding spaces, what the error bounds are for recovering the correct latents in non-ideal conditions, and how the performance degrades as we move further from the assumptions of the proof in practice to justify the ensembling and alignment step..
- Related, aligning N-1 models to an anchor model requires training N-1 separate alignment networks. Each alignment process involves optimizing over the entire dataset. This process scales linearly with the number of models in the ensemble, which could become prohibitively expensive for large ensembles. The alignment process adds an additional training phase after the initial pre-training of the models. For large datasets or complex models, this could significantly increase the overall time required to deploy the ensemble alignment method. If the alignment to the anchor model is imperfect, these errors could propagate through the entire ensemble (e.g. outliers that sit far away from other models' vectors). Given that, the choice of anchor model could introduce bias, even by a random pick. It would be valuable to see the authors' response on that, or provide some ways of evaluating the alignment.

---

### Official Review · Reviewer_hqVt · 2024-10-06
**This submission studies the generalization of pre-trained representations to out-of-distribution (OOD) data in the context of self-supervised learning. The authors propose Ensemble-InfoNCE, a method designed to align ensembles obtained from a pre-trained encoder to improve generalization on OOD data while maintaining strong performance on in-distribution data. Preliminary experiments on the MNIST dataset show that the aligned embedding-space ensemble significantly enhances the quality of OOD embeddings compared to baseline models.**

**Rating:** 8
**Confidence:** 4

**Review:**

**Strengths:**
- Clear and well-defined problem statement, along with a thorough discussion of the current challenges in existing works with unlabeled data.
- Strong explanation of key concepts.
- Promising results demonstrated in the experiments.

**Comments:**
The clarity of the paper could be improved by providing more detailed mathematical explanations, particularly on how the learned $R$ is applied within the Karcher Mean algorithm. Including this will help readers better understand the process of the proposed method.

**Limitations:**
A notable limitation of the method is its potential computational cost, especially when different $R_j$ matrices need to be learned for all $j$ pre-trained encoders, which could become infeasible with large datasets. To address this concern, the authors should consider including an ablation study that examines the impact of the number of different $R_j$ matrices to be learned. This would offer valuable insights into how the method scales and help determine an optimal trade-off between performance and computational efficiency.

---

### Official Review · Reviewer_owvP · 2024-10-07
**The paper offers an interesting theoretical contribution, but the limited experimental scope and reliance on MNIST reduce its overall impact.**

**Rating:** 5
**Confidence:** 4

**Review:**

Review:
The paper proposed a approach to improving the generalizability of pre-trained encoders on out-of-distribution (OOD) data using embedding-space ensembles. The authors' method, termed Ensemble-InfoNCE, aligns the embedding spaces of self-supervised pre-trained encoders to enhance the quality of embeddings on both in-distribution (ID) and OOD data. The approach offers a balance between interpretability and flexibility by combining the advantages of predictive-space and weight-space ensembling.

Quality:

The paper is well-structured, and the methodology is clearly presented. The theoretical analysis, particularly around the orthogonality relationship between embedding spaces, is sound and provides a strong basis for the proposed unsupervised alignment method. However, the experimental evaluation is limited, and more diverse datasets beyond MNIST are necessary to fully establish the validity of the approach across more complex OOD scenarios.

Clarity:

The clarity of the paper is generally good, with a clear explanation of the theoretical background and the methodology.

Originality:

The idea of performing ensembling in the embedding space with an alignment mechanism is novel and may contributes to the field of self-supervised learning.

Significance:

The paper addresses an important issue in the field of machine learning—generalizing pre-trained encoders to OOD data. While the contribution is valuable, the significance is somewhat limited by the narrow scope of the experiments. The reliance on a single dataset (MNIST) and limited evaluation metrics (Recall@1 and MAP@R) reduce the impact of the findings, as generalization to larger and more complex datasets is yet to be demonstrated.

Pros:

Novel method combining the benefits of both predictive-space and weight-space ensembling.
Improvements in OOD generalization on the MNIST dataset compared to single encoders.

Cons:

Evaluation is conducted only on a single, relatively simple dataset (MNIST).
The paper only uses two metrics (Recall@1 and MAP@R), which may not fully capture the performance.
Limited discussion on the applicability of the approach to larger, real-world datasets.

---

### Decision · Program_Chairs · 2024-10-10

**Decision:**

Accept

**Comment:**

In light of the positive reviewers' feedback and relevancy of the submission, we are pleased to accept this paper for presentation at UniReps 2024. We kindly ask the authors to incorporate the reviewers' suggestions and feedback in the final camera-ready version of the manuscript.